# Nondestructive Monitoring of Soft Bottom Fish and Habitats Using a Standardized, Remote and Unbaited 360° Video Sampling Method

Delphine Mallet [1], Marion Olivry [2], Sophia Ighiouer [3], Michel Kulbicki [4] and Laurent Wantiez [5,*]

1 VISIOON, 6 Rue du Docteur Fruitet, 98800 Nouméa, New Caledonia; d.mallet@visioon.nc
2 CNAM Intechmer, Boulevard de Collignon, 50110 Tourlaville, France; marionolivry4@gmail.com
3 École Pratique des Hautes Études, PSL University, Les Patios Saint-Jacques, 4-14 Rue Ferrus, 75014 Paris, France; sophia.ighiouer@hotmail.fr
4 Institut de Recherche pour le Développement (IRD), UMR Entropie, Université de Perpignan, 66000 Perpignan, France; michel.kulbicki@ird.fr
5 Entropie, University of New Caledonia, 98851 Nouméa, New Caledonia
* Correspondence: laurent.wantiez@unc.nc

**Abstract:** Lagoon soft-bottoms are key habitats within coral reef seascapes. Coral reef fish use these habitats as nurseries, feeding grounds and transit areas. At present, most soft-bottom sampling methods are destructive (trawling, longlining, hook and line). We developed a remote, unbaited 360° video sampling method (RUV360) to monitor fish species assemblages in soft bottoms. A low-cost, high-definition camera enclosed in a waterproof housing and fixed on a tripod was set on the sea floor in New Caledonia from a boat. Then, 534 videos were recorded to assess the efficiency of the RUV360. The technique was successful in sampling bare soft-bottoms, seagrass beds, macroalgae meadows and mixed soft-bottoms. It is easy to use and particularly efficient, i.e., 88% of the stations were sampled successfully. We observed 10,007 fish belonging to 172 species, including 45 species targeted by fishermen in New Caledonia, as well as many key species. The results are consistent with the known characteristics of the lagoon soft bottom fish assemblages of New Caledonia. We provide future users with general recommendations and reference plots to estimate the proportion of the theoretical total species richness sampled, according to the number of stations or the duration of the footage.

**Keywords:** underwater video; ichthyofauna; seagrass bed; macroalgae soft substrate; perireefal





## 1. Introduction

Soft bottom habitats constitute a major part of the coral reef seascape. These habitats make up extensive areas of mud, sand or rubble that marine plants can colonize [1,2]. In a lagoon environment, they are the key corridors between coral reefs, playing an essential role in ensuring connectivity and energy transfer within a mosaic of reef and perireefal habitats [3,4]. Many fish species, along with several emblematic species such as sea turtles or dugongs, use these habitats. Fish use such habitats as nurseries, feeding grounds or transit areas [5,6]. This very complex seascape is under increasing anthropogenic pressure, in particular due to the growing population and increased impacts such as fishing, coastal development, tourism, inputs from watersheds, the transformation of coastal landscapes and marine aquaculture.

Few studies have been devoted to soft bottom habitats compared to the other ecosystems of this seascape such as coral reefs or mangroves [7]. One of the main reasons for this is that soft bottom fish assemblages are difficult to sample, as individuals are scattered over very large areas and are often associated with significant depths. Most of the available data come from experimental fishing (essentially hook and line or trawl) or fish landings

(e.g., [7–11]), which are extractive methods and present the typical problems of representativeness, sensitivity and repeatability. While standardized and nondestructive sampling methods such as underwater visual censuses (UVC) are extensively used on coral reefs, these approaches are not adapted to soft bottom habitats because of the low occurrence of fish, specific fish behavior as well as the extent or the depth of these habitats. In New Caledonia, soft bottom fish assemblages are, at present, known only from earlier programs based on experimental catch data [11–16] and fisheries survey data [17–19].

The recent development of underwater video systems [20,21] provides an opportunity to develop a standardized method to monitor fish assemblages over large areas such as soft bottom habitats. This tool has the advantage of being nondestructive for the environment, has little influence on fish behavior, and can record for long periods at various frequencies. Different video techniques exist to sample fish such as remote underwater video, whether baited or not, diver-operated video or towed video (see [21] for a review on video techniques). At present, the most widely used approach in perireefal habitats is the 'BRUV' technique (Baited Remote Underwater Video) that attracts fish around the camera with bait (see [22–25] for applications on seagrass beds). In New Caledonia, video systems have been mainly used in censuses of coral reef fish [26–28] or sharks [29,30]. Pelletier et al. (2012) used video techniques on soft bottom habitats, but the performance of the method (required number and duration of videos) was not tested.

Pilot studies on method efficiency are important to validate and optimize sampling methods as part of developing cost effective and statistically robust monitoring programs. However, most sampling designs based on video techniques are used without such pilot studies, which may compromise their results [31–33]. Considerable variability in sampling times and number of replicates characterize published studies [21]. Recently, Garcia et al. (2021) studied the possible trade-off between the number and the length of remote videos used in a rapid assessments of reef fish assemblages. With 46 videos on five sites, they indicated that increasing the sampling coverage in the reef area may be more effective than just extending the video length.

The objective of this study was to perform a pilot study to present a standardized sampling protocol to monitor the diversity, abundance and structure of perireefal fish assemblages during daytime, in relation to the environment. We used a remote and unbaited 360° video system (RUV360). The 360° camera records simultaneously all the area around each sampling point. The aim of this pilot study was to assess: (1) the limits of the RUV360 sampling method (cost, visibility, current, bottom topography); (2) the fish species targeted by the technique; and (3) the optimal recording time per station and the number of stations required to obtain representative, stable and reproducible data on perireefal fish communities.

## 2. Material and Methods

### 2.1. Study Area and Sampling Design

The main island of New Caledonia is one of the largest coral reef lagoons in the world (19.385 km$^2$). It includes 16,874 km$^2$ of nonreef habitat, with certain areas listed as a UNESCO World Heritage site [34]. This very complex seascape is under increasing anthropogenic pressure, in particular due to the growing population (268,767 inhabitants in 2014 compared to 230,789 in 2004; www.isee.nc (accessed in 2019)), and increased impacts such as fishing, coastal development, tourism, mining and marine aquaculture. The study was conducted from the 3 May until the 18 July 2018, in the Southwest Lagoon of New Caledonia. The study area is an 18.5 km long and 4 km wide transect from the coastline to the barrier reef (Figure 1). This area is representative of the coral reef seascape of the Main Island, near Nouméa, the capital city. The lagoon includes 67.5 km$^2$ of soft bottom habitats and two rows of coral reefs and coralline islets along a shore-barrier reef gradient. Coral heads are scattered on the lagoon bottom. Habitats with more than 50% hard substrate were excluded from the sampling.

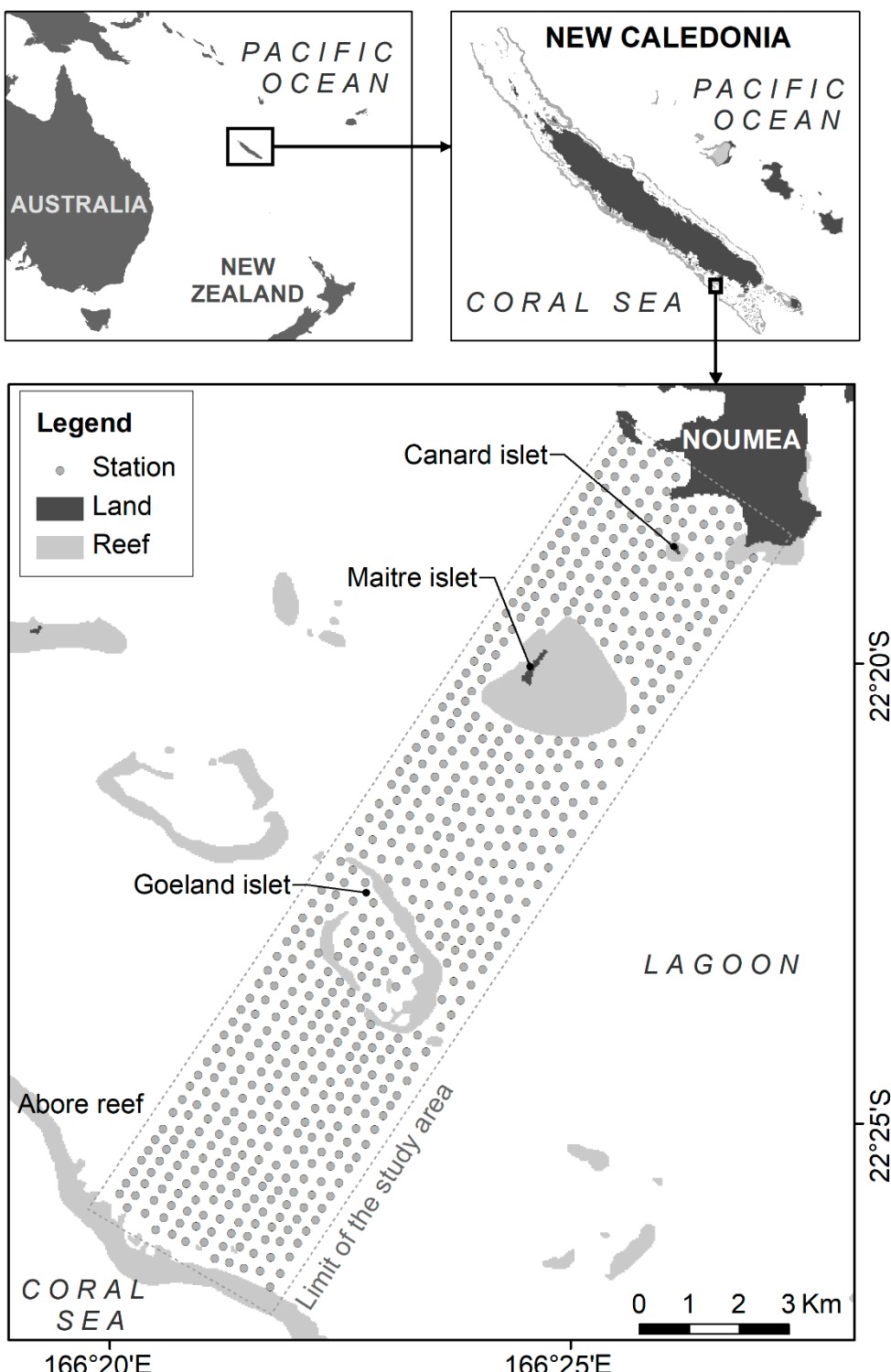

**Figure 1.** Studied area and sampling design. Each dot represents a station.

　　In order to assess the optimal recording time for each station and the number of stations required to get representative and reproducible data on soft bottom fish assemblages, we had to oversample the area. A systematic sampling protocol including 609 stations within a grid of 300 m wide cells was used. The distance between stations was sufficient to avoid overlap due to fish swimming from one station to another. The stations were sampled during daylight, at least one hour after sunrise and one hour before sunset, to avoid possible crepuscular variation in fish assemblages [35].

### 2.2. Sampling Technique and Images Analysis

This study used an autonomous, remote and unbaited video technique named "RUV360" (Figure 2). The camera was a low-cost camera (250€) from KODAK (model PIXPRO SP360 4K) which can record videos in very high definition (1440 × 1440 pixels, 30 fps), featuring a spherical lens with a 360° horizontal and a 235° vertical view, pointed directly upward (Figure 2). The camera was enclosed in a waterproof housing (limited to 60 m depth, 50€), fixed to an aluminum tube 17 cm above the seafloor. A tripod system was used to position and stabilize the camera on the sea floor (Figure 2). The video system was deployed from a boat without the need for the crew to enter the water. This method allowed us to maximize the number of observations while minimizing disturbance to the environment (no boat and no human were present near the video system during the recordings). To evaluate the minimum recording duration required to have representative observations, we fixed the recording duration at 25 min. This time was sufficient to observe sedentary fish and then assess the amount of additional information (passing fish) obtained over time.

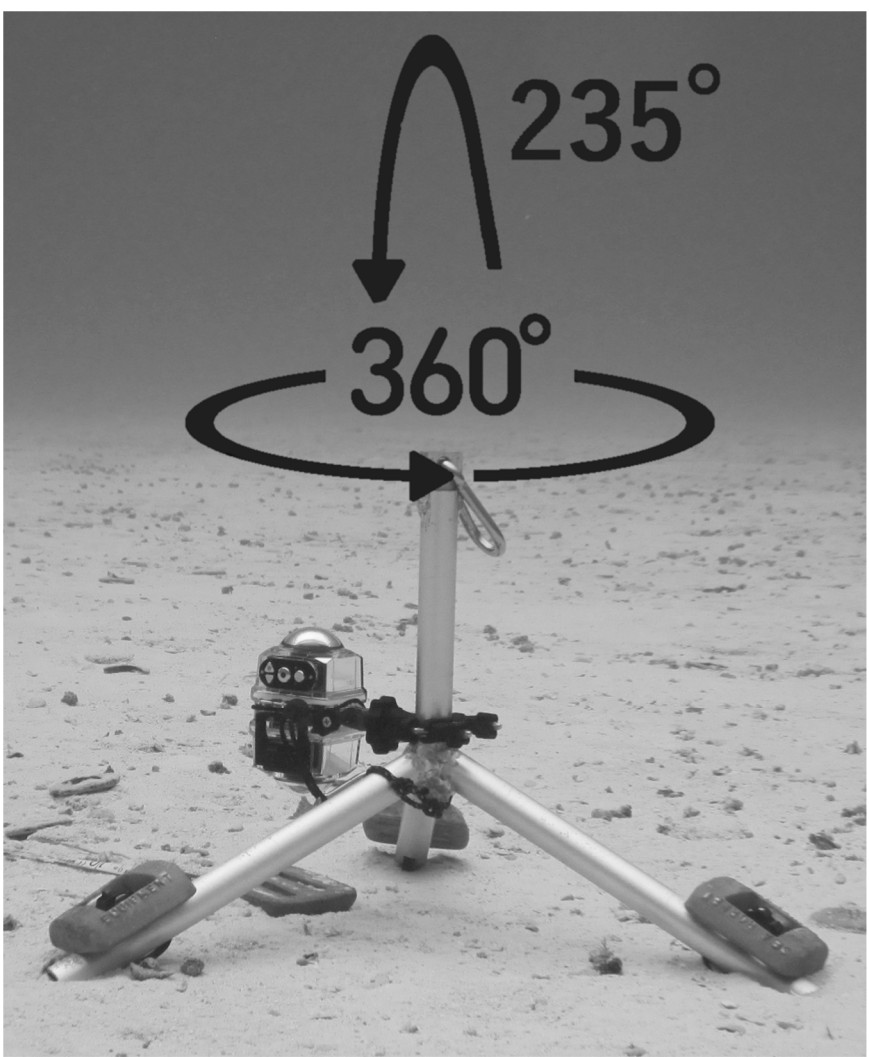

**Figure 2.** Picture of the remote underwater video system on the seabed.

To optimize sampling at sea, we used four video systems deployed by two people aboard a small boat (<8 m). After each sampling day at sea, all videos were checked to assess (1) an appropriate field of view (visibility > 5 m), (2) an appropriate orientation of the camera allowing for a clear view of the seabed, (3) a stable camera during filming, and (4) that the habitat sampled was mainly soft (<50% of hard bottom). When a video was found to be invalid, a second attempt was made the following day.

All videos were analyzed by the same experienced observer using the camera software (Kodak Pixpro SP360 PC software, v1.7.0). The habitat was characterized by estimating the percentage of abiotic and biotic coverage over the 360° images using the "MSA" protocol [36]. The abiotic cover was classified as bare sediment (mud, sand, gravel and small boulders < 30 cm) or nonliving hard substrate (dead corals, coral slab, blocks > 30 cm). The biotic cover (live substrate) was classified as live corals (carbonated edifices that were still in place and present a coral shape) or "marine plants" (seagrass and macroalgae). The videos did not allow us to differentiate systematically between seagrass (*Cymodocea* sp., *Halophila* sp., *Halodule* sp., *Syringodium* sp., *Thalassia* sp.) and macroalgae (*Caulerpa* sp., *Halimeda* sp., *Lobophora* sp., *Sargassum* sp., *Turbinaria* sp.). All fish were counted and identified at the lowest possible taxonomic level. To avoid counting the same fish several times, we used a conservative measure of relative abundance: "MaxN" [37]. This measure of abundance is the maximum number of individuals of the same species appearing at the same time throughout the entire video. To study the influence of camera soak time on species composition and abundances, we calculated MaxN (by species) every 30 sec. This protocol made it possible to study the number of new species and new individuals observed within each time interval. Some fish species from the same genus are similar and only differ by small details (eye color, small color dots, etc.). These species are therefore difficult to distinguish on videos unless they are close enough to the camera. For our video analyses, we aggregated these species into groups: (i) *Amphiprion* gp for *Amphiprion akindynos* and *Amphiprion clarkii*; (ii) *Lethrinus* gp for *Lethrinus variegatus* and *Lethrinus genivittatus*; (iii) *Nemipterus* gp for *Nemipterus peronii*, *Nemipterus furcosus* and *Nemipterus zysron*; (iv) *Parapercis* gp for *Parapercis australis* and *Parapercis millepunctata*, and (v) *Pomacentrus* gp for *Pomacentrus amboinensis* and *Pomacentrus moluccensis*.

### 2.3. Sampling Cost

We estimated the cost of sampling by the time required for fieldwork and video analysis. The total time required for fieldwork each day included preparing the boat, the trip to the sampling area and the time spent within the sampling area (setup of the video systems, deployment and retrieval of video systems, travel between stations). The time required to characterize the habitat, then identify and count the macrofauna on the videos, was noted for each video during the video analysis.

### 2.4. Data Analysis

2.4.1. Typology of the Habitat and Fish Assemblages

We selected all stations composed of less than 50% hard bottoms for our study on soft bottom habitats. To identify the typology of the habitat, we performed a principal component analysis (PCA) on raw data and a hierarchic ascending classification (HAC) on the first three axes of the PCA (100% of the inertia), using the squared Euclidean distance and Ward's aggregation method [38].

In order to verify the discriminating nature of the type of soft bottom on the fish assemblages, a CAP (canonical analysis of principal coordinates) was carried out on the Bray Curtis similarity matrix between the stations according to species abundance, using habitat type as a classifier. We applied a square root transformation on the dataset prior to analysis to down-weight the importance of the outlier species [39]. The results of the CAP were validated by a PERMANOVA (999 permutations).

2.4.2. Influence of Soak Time and Number of Stations Sampled on Fish Assemblages

The relationship between soak time and the number of species or individuals recorded was modelled using species accumulation curves and cumulative abundance curves. Species richness and abundance were calculated at 30 s intervals until the 25 min of soak time elapsed, for the entire area and per habitat.

The species accumulation models used a rarefaction method based on raw data added in ascending order. The rarefaction model known as Mao Tau's estimate [40] is a powerful

tool for detecting species richness [41]. Abundance accumulation models used the time required to reach MaxN at each station added in ascending order. The estimate of the theoretical total number of species or individuals in the area studied was calculated by fitting a nonlinear Michaelis-Menten model [42] (the most accurate of the models tested) to the accumulation data: $y = (Vm \times t)/(K + t)$, where y is the number of species or individuals after t min of recording, "Vm" is the theoretical total number of species or individuals in the study area, and "K" is the number of stations where half of the theoretical total number of species or individuals have been detected in the videos.

We calculated the proportion of the theoretical species richness (SR) according to the number of stations and the duration of the footage. These proportions were calculated as the average of the SR obtained by 180 s intervals using 999 draws (without replacement) of the required number of stations in the overall data set (534 stations).

## 3. Results

### 3.1. Sampling Cost

We validated 534 stations out of the 609 stations of the sampling protocol in the area, between 1 and 25 m depth (mean ± SE = 12.9 ± 0.3 m). Fifty stations, located in a coral habitat (more than 50% of live coral), were excluded from the study. It was not possible to position the camera correctly at 58 stations due to the relief of the seabed. The visibility of the water was too low for 8 stations and the current was too high for 52 stations (especially in the channels). Depending on wind, wave and depth conditions, the preparation of the boat and the trips took between 19 min and 113 min (mean ± SE = 46 min ± 3 min) (Table 1). Each 25 min of video required an additional of 10 min to set up, deploy and retrieve the video system. This time was reduced by using four RUV360 systems simultaneously, resulting in a total time of 40 min to 92 min to sample a set of four stations (mean ± SE = 40 min ± 3 min). The variations in time are mostly due to the requirement of correct positioning of the system on the seabed (depending on the percentage of hard corals, the depth and the relief) and the distance between two stations. The analyses of the 534 videos took 425 h in all. The time to analyze one video was between 24 min and 78 min (mean ± SE = 49 min ± 3 min), depending on the complexity (number of species and abundances) of the biodiversity in the video.

**Table 1.** Sampling cost. Min, max and mean (± SE) correspond to the time required per day in minutes for fieldwork preparation, per set of four stations and per station for video analysis. Totals correspond to the time required to sample and analyze the 534 videos of the study.

| Time Required (Min) | Fieldwork | | Analysis of One Video |
|---|---|---|---|
| | Daily Preparation of Boat and Material + Trips to the Sampling Area | Sampling a Set of 4 Stations | |
| Min | 19 | 40 | 24 |
| Max | 113 | 92 | 78 |
| Mean ± SE | 46 ± 3 | 40 ± 3 | 49 ± 3 |
| Total | 1123 | 7839 | 25,494 |

### 3.2. Typology of the Habitat

The stations were mainly composed of bare sediment and marine plants. Overall, 31 stations were almost exclusively composed of bare sediment (more than 90% of the habitat), and 66 were almost exclusively composed of marine plants (more than 90% of the habitat); 119 stations had living corals, which never exceeded 35%, and nonliving hard substrate (max 20%) was present at 52 stations.

It was possible to identify three habitats in the studied area (Figure 3). The "vegetated soft bottom habitat" (317 stations) was dominated by marine plants (from 52% to 100%) and very little hard substrate (from 0% to 10% of living corals and from 0% to 5% of nonliving

hard substrates). The "bare soft bottom habitat" (160 stations) was dominated by bare sediments (from 50% to 100%), very little hard substrate (from 0% to 10% of living corals and from 0% to 5% of nonliving hard substrates) and a lower percentage of marine plants (from 0% to 50%). The "mixed soft bottom habitat" (57 stations) was characterized by hard substrate (from 10% to 40%), including nonliving hard substrate (from 0% to 20%) and/or scattered living corals (from 0% to 35%).

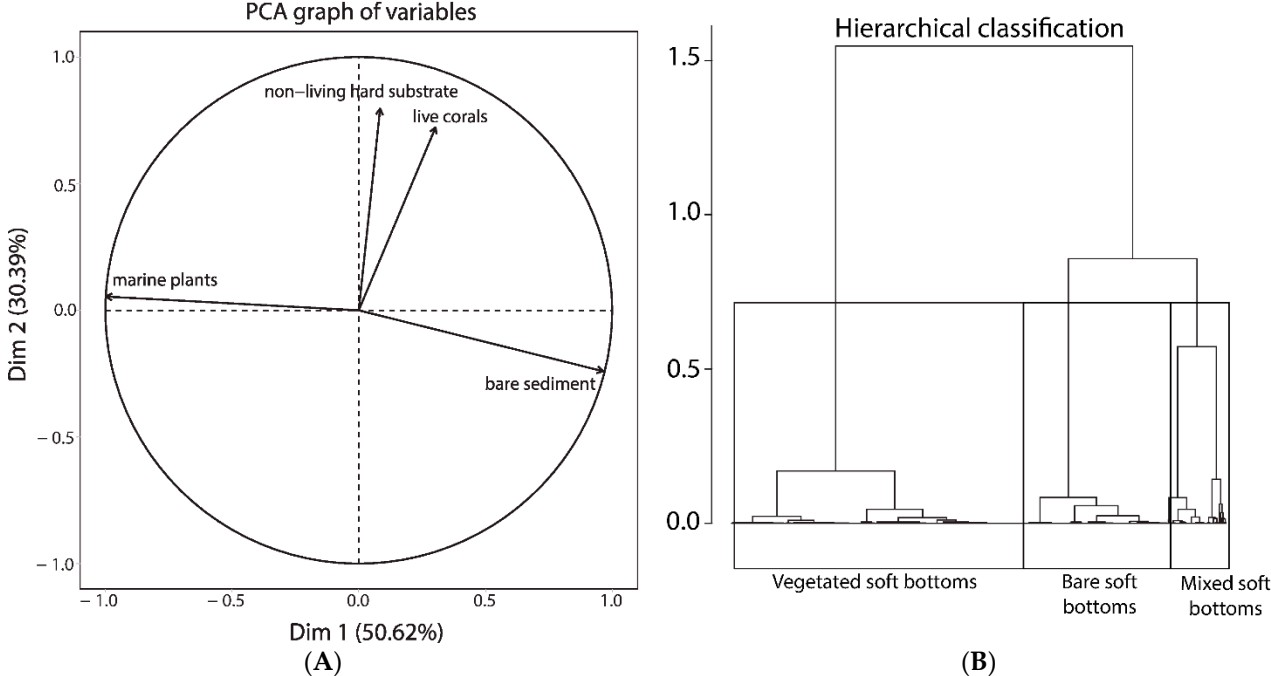

**Figure 3.** PCA of the habitats characteristics per station (**A**) and typology of the habitat (**B**).

### 3.3. Fish Assemblages

In all, 10,007 fish belonging to 172 species (98 genera and 37 families) were observed (Supplementary Materials Table S1); 3534 fish (26% of the fish) observed at 330 stations (62% of the stations) could not be identified, because they were too small (1774-50%), were located in the upper water column (607-17%) or were at the limit of detectability (506-14%). The rest of the unidentified fish showed no distinctive signs (361-10%), were blurred (260-8%) or swam too quickly (26-1%) to be identified.

Among the fish identified, the most frequent and abundant families were the Lethrinidae (frequency of occurrence (freq) = 33%, MaxN summed across all deployments (total MaxN) = 992), Pomacentridae (freq = 26.8%, total MaxN = 3390), Labridae (freq = 26.4%, total MaxN = 1175) and Mullidae (freq = 25.8%, total MaxN = 811). Most of the species were carnivores (99 species, 4184 fish). Plankton feeders were second in terms of MaxN (3866 fish), but were also the least diverse (18 species) (Table 2).

**Table 2.** Number of families, genera, species and abundance of fish (MaxN) per trophic group.

| Trophic Group | Families | Genera | Species | MaxN |
|---|---|---|---|---|
| Carnivores | 22 | 58 | 99 | 4184 |
| Herbivores-detritus | 7 | 14 | 29 | 1507 |
| Piscivores | 7 | 18 | 26 | 450 |
| Plankton feeders | 7 | 12 | 18 | 3866 |

On average, the video recorded 4.1 species and 19 fish per station for the full 25 min of deployment (Table 3). There were important variations between stations, from no fish at 119 stations to a maximum of 28 species and 269 fish at one station. Commercial species

made up 29% of the fish species per station and 33% of the individuals per station. The most diverse (34% of the commercial species) and abundant (30% of the MaxN of the commercial fish) commercial fish were Lethrinidae. Scaridae (21% of the species and 23% of the MaxN of commercial fish), Carangidae (11% of the species and 15% of the MaxN of commercial fish) and Acanthuridae (11% of the species and 8% of the MaxN of commercial fish) followed in order of importance.

**Table 3.** Mean specific richness and abundance per station ($\pm$SE) for all the ichthyofauna, for the commercial species and for the 4 more frequent commercial families.

|  | Species Richness per Station | Abundance per Station (MaxN) |
|---|---|---|
| Total ichthyofauna | $4.1 \pm 0.2$ | $19.0 \pm 1.4$ |
| Commercial species | $1.2 \pm 0.1$ | $6.3 \pm 0.6$ |
| Lethrinidae | $0.41 \pm 0.03$ | $1.86 \pm 0.20$ |
| Scaridae | $0.25 \pm 0.03$ | $1.44 \pm 0.24$ |
| Carangidae | $0.13 \pm 0.02$ | $0.95 \pm 0.34$ |
| Acanthuridae | $0.13 \pm 0.02$ | $0.53 \pm 0.15$ |

The fish species richness and MaxN were significantly influenced by habitat (PERMANOVA, $p = 0.001$). Species richness and MaxN were higher in the mixed soft bottom habitat than in the bare or vegetated soft bottom habitats (paired comparisons, $p < 0.001$). The fish assemblages were different on the three soft bottom habitats (PERMANOVA, $p = 0.001$). A canonical analysis was carried out on the first 42 axes of the analysis in principal coordinates (98.54% of the total inertia) (Figure 4). The CAP was validated ($p = 0.001$) and indicated an overall percentage of correct and stable classification of 63%. First, the model discriminated mixed soft bottoms communities (88% correct classification). The discrimination of the assemblages in the two other habitats was lower, i.e., 59% on the vegetated soft bottoms and 59% on the bare soft bottoms. These assemblages shared more similarities (75% misclassification between them). The mixed soft bottom fish assemblage was the most diverse. This assemblage was characterized by the presence of hard bottom species associated with corals, such as damselfish (*Dascyllus aruanus* and unidentified damselfishes), butterfly fish (*Chaetodon mertensii*), angelfish (*Centropyge tibicen*), parrotfish (*Chlorurus sordidus*, *Scarus schlegeli* and unidentified parrotfish), one wrasse (*Thalassoma lunare*), one triggerfish (*Sufflamen chrysopterum*) and coral trout (*Plectropomus leopardus*). Several ubiquitous species also characterized this community, such as goatfish (*Parupeneus barberinoides*, *Parupeneus multifasciatus*) and sea bream (*Gymnocranius* sp.). The presence of species associated with seagrass beds or algae meadows characterized the vegetated soft bottom fish assemblage, in particular two emperors (*Lethrinus variegatus* and *Lethrinus genivittatus*), one leather jacket (*Paramonacanthus japonicus*) and two wrasses (*Oxycheilinus bimaculatus* and *Suezichthys devisi*). The bare soft bottom fish assemblage was the least diverse. Its main characteristic was the absence of hard bottom species or vegetated soft bottom species. The only fish observed on these bottoms were specimens moving between the other habitats of the lagoon. However, this assemblage was characterized by the presence of spangled emperors (*Lethrinus nebulosus*) which frequent the large areas of the lagoon with a preference for sandy bottoms, where they find their food.

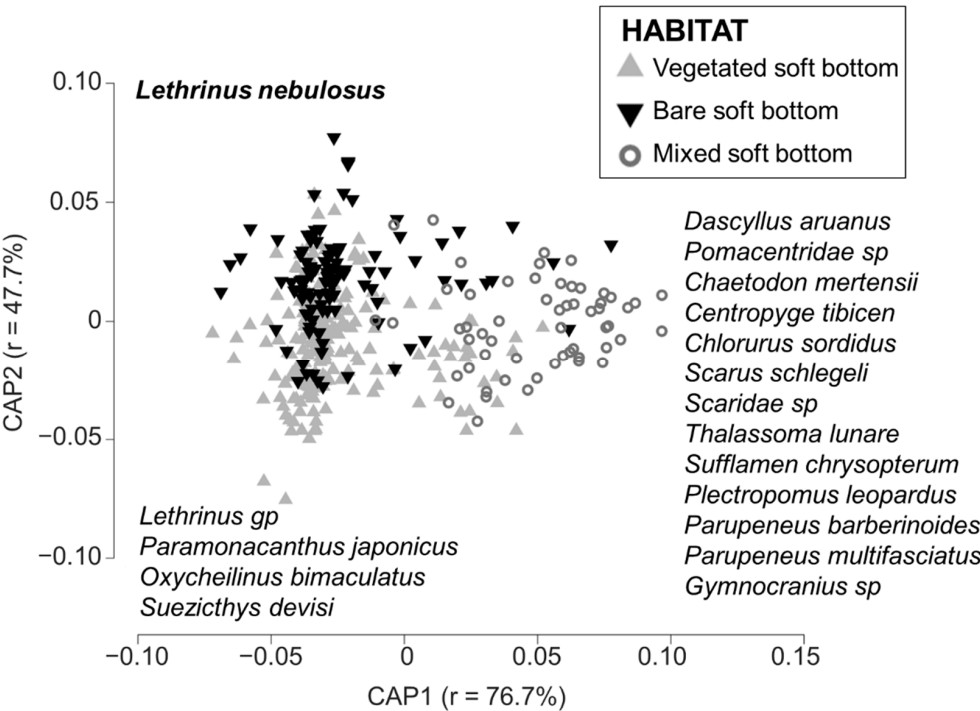

**Figure 4.** CAP of fish assemblage between stations, under constraint of habitat type. Species with a correlation $\geq 0.35$ to the first factorial design are specified for each habitat.

### 3.4. Influence of Soak Time and Number of Stations Sampled on Fish Assemblages

The deployment duration had a significant effect on the species richness (SR) and abundance (MaxN) observed by station (Friedman test, $p < 0.001$). The average number of species observed per station increased from $1.2 \pm 0.6$ (SR $\pm$ SE) species with 30 sec of observation to $4.2 \pm 1.5$ species with 25 min (Figure 5A). The SR was stable after 7.5 min of observation (pairwise comparisons test, $p > 0.05$). The MaxN per station also increased significantly over time (MaxN $\pm$ SE = $8.0 \pm 6.6$ fish after 30 sec and $18.8 \pm 10.5$ fish after 25 min) (Friedman test, $p < 0.001$). The MaxN was stable after 1.5 min (pairwise comparisons test, $p > 0.05$). The SR increased very quickly over time at the beginning of the recording (Figure 5B), before dropping progressively to reach an asymptote corresponding to the total theoretical species richness according to the footage duration (Michaelis-Menten model, theoretical SR-time = 173 species) in the study area: 80% of the theoretical SR-time was observed after 5 min and 95% of the theoretical SR-time after 14 min (Table 4). The theoretical SR-time was not significantly different between habitats (Chi-squared test, $p > 0.05$). Within the vegetated and mixed soft bottoms, SR progressed very quickly at the beginning of the recordings: 80% of the theoretical SR-time was observed after 5 min on the vegetated soft bottoms and 4.5 min on the mixed soft bottoms (Table 4). In contrast, the SR on the bare soft bottoms increased more slowly at the beginning of the recordings: 11 min were necessary to observe 80% of the theoretical SR-time on this habitat. However, 95% of the theoretical SR-time on the bare soft bottoms was observed within 17 min, which was only 1.5 to 2.5 min more than for the other soft bottom habitats.

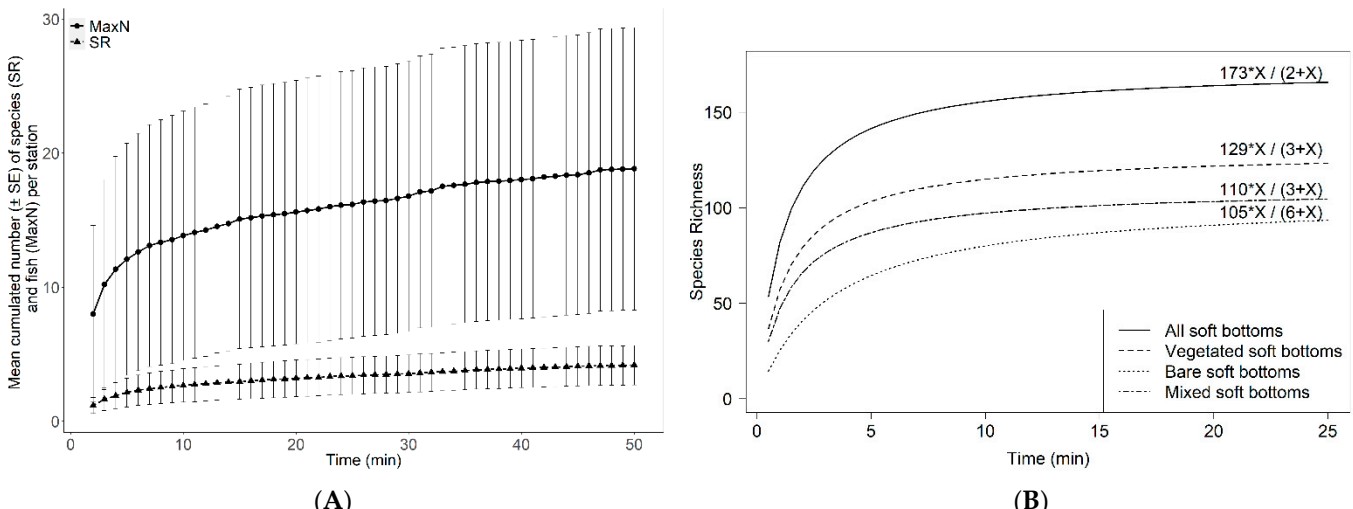

**Figure 5.** (**A**) Mean cumulated number (±SE) of species and fish per station in 30 sec time intervals up to 25 min. (**B**) Species accumulation curves in 30 s time intervals up to 25 min for all soft bottoms and by habitat (see legend). Equation of the curve for each habitat is given on the corresponding curves.

**Table 4.** Deployment duration necessary to observe 50, 80, 85, 90, 95% of the theoretical SR-time. Deployment durations were evaluated from the accumulation curves calculated as a function of time over all the stations and by habitat.

| Proportion of the Theoretical SR-Time (%) | Deployment Duration | | | |
|---|---|---|---|---|
| | **All Soft Bottoms** | **Bare Soft Bottoms** | **Vegetated Soft Bottoms** | **Mixed Soft Bottoms** |
| 50 | 1 min 06 | 3 min 15 | 1 min 15 | 1 min 18 |
| 80 | 5 min 00 | 11 min 00 | 5 min 00 | 4 min 30 |
| 85 | 7 min 00 | 14 min 00 | 9 min 00 | 7 min 30 |
| 90 | 10 min 00 | 15 min 30 | 11 min 30 | 10 min 30 |
| 95 | 14 min 00 | 17 min 00 | 14 min 30 | 15 min 30 |

There was no significant link between the number of stations and the estimates of SR or MaxN observed per station (Spearman correlation, $p > 0.05$). Indeed, the mean number of species observed per station was relatively stable, regardless of the number of stations sampled. It varied from 3.9 species on average per station with 2 stations to 4.1 species on average per station with 534 stations (Supplementary Materials Figure S1). On the other hand, the standard error (SE) decreased significantly as the number of stations increased (SE for 2 stations = 2.5 and SE for 534 stations = 0.2). The average abundance (MaxN) per station followed the same trend. It was relatively stable and ranged, on average, from 19.0 fish per station for 2 stations to 18.9 fish per station for 534 stations (Supplementary Materials Figure S1). The SE of relative abundance per station also decreased significantly as the number of stations increased (SE for 2 stations = 13.4 and SE for 534 stations = 1.4). The SR gradually increased depending on the number of stations sampled (Figure 6). The total theoretical SR according to the stations sampled (theoretical SR-station) estimated by the model within the study area was 195 species. Eighty percent of the theoretical SR-station was observed with 369 stations (6.2 stations/km$^2$ in the study area), while 88% was observed with all the stations sampled (534 stations or 9 stations/km$^2$) (Table 5). The theoretical SR-station was not significantly different between habitats (Chi-squared test, $p > 0.05$). Within the vegetated and mixed soft bottoms, SR progressed more quickly than for bare soft bottoms: 80% of the theoretical SR-station was observed with 265 stations (7.5 stations/km$^2$) on the vegetated soft bottoms and 70 stations (11.1 stations/km$^2$) on the mixed soft bottoms (Table 5). Again, in contrast, the SR on the bare soft bottoms increased more slowly depending of the number of stations sampled: 320 stations (17.9 stations/km$^2$) were necessary to observe 80% of the theoretical SR-station on this habitat.

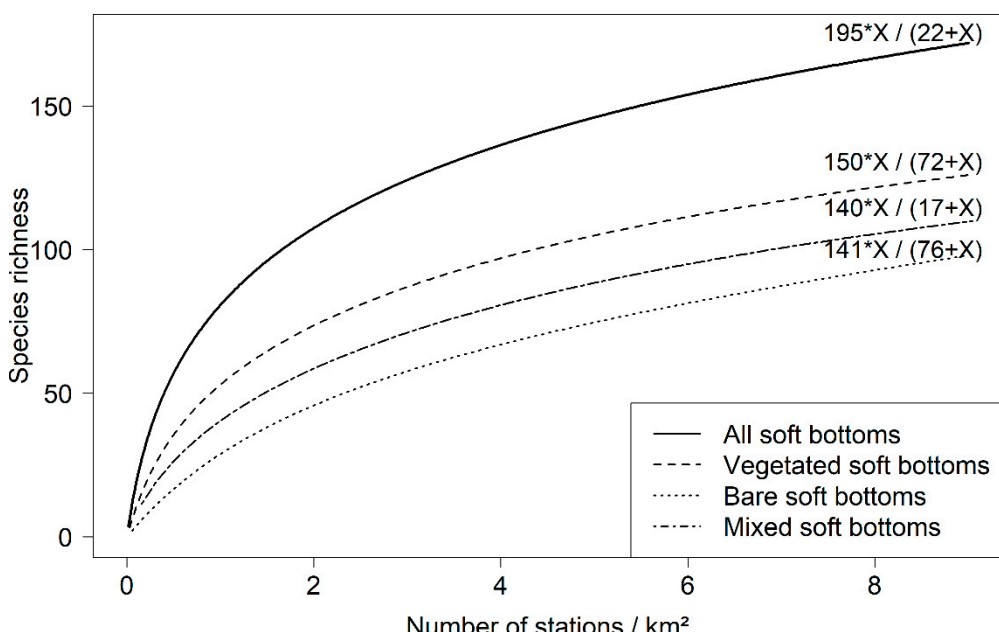

**Figure 6.** Species accumulation curves per number of stations/km² for all soft bottoms and by habitat (see legend). Equation of the curve for each habitat is given on the corresponding curves.

**Table 5.** Number of stations per km² required to observe 50, 80, 85, 90 and 95% of the theoretical SR-Scheme 25. min.

| Proportion of the Theoretical SR-Station. (%) | Number of Stations per km² | | | |
| --- | --- | --- | --- | --- |
| | **All Soft Bottoms** | **Bare Soft Bottoms** | **Vegetated Soft Bottoms** | **Mixed Soft Bottoms** |
| 50 | 1.6 | 4.3 | 2 | 2.7 |
| 80 | 6.2 | 18 | 7.5 | 11.1 |
| 85 | 7.8 | 24.4 | 11.5 | 15.9 |
| 90 | 14.1 | 38.8 | 18.2 | 24.6 |
| 95 | 29.5 | 83.7 | 38.4 | 55.6 |

## 4. Discussion

An unbaited video technique was selected because it did not attract fish to the camera. Using bait to attract fish would modify the fish assemblage because fish species react differently to bait [21,33,43,44]. The objective was to get a less biased representation of the assemblage during daytime. A 360° video technique was selected to sample in all directions simultaneously and record all fish in the sampling area.

### 4.1. Fieldwork Implementation and Costs

The RUV360 was easy to use and particularly efficient, since 88% of the initially selected stations were successfully sampled. The approach appears to be more efficient than other unbaited, multidirectional video systems. For instance, the "STAVIRO" (rotating video system), described by Pelletier et al. [26], for use on hard- and soft-substrate habitats successfully sampled 70% of the stations during a pilot study and reached 81% validation in a subsequent studies. More recently, the "compact video lander" developed by Watson and Huntington [45] was used on rocky reefs, and successfully sampled 70% of the stations. When deploying video systems directly from a boat, one of the main causes of nonvalidation is an inappropriate orientation of the camera towards the seabed. Only 3.1% of nonvalidated stations of the present study were attributed to seafloor relief issues. Such problems were reduced with the RUV360 because (1) we targeted only soft bottoms which are less complex than hard substrate, and (2) the camera had a 235° vertical field of view (V-FOV), compared to cameras generally used in other video techniques (BRUV, RUV or

STAVIRO), that have a V-FOV of 60° for the wide angle lens of the latest Sony (specification of the model FDR-AX700 on www.sony.com) and 94.4° for the wide angle lens of the latest Gopro (specification of the model HERO8 Black on www.gopro.com). The RUV360 was also efficient in terms of other typical causes of nonvalidation. It was particularly stable (only 0.3% of nonvalidated stations were attributed to its instability) and could be used in channels where tide currents occurred. The impact of a low visibility was limited because the FOV was good (only 0.7% of nonvalidated stations were attributed to the visibility).

The cost associated with the use of the RUV360 method was evaluated as a combination of the time required for sampling and video analyses. Fieldwork was estimated for the simultaneous deployment of four RUV360s within a systematic sampling grid of 300 m width and a 25 min video recordings per station using one boat (<8 m, two persons minimum). The RUV360 appears to be an efficient alternative to other video systems, although comparisons are complicated, as very few studies provided cost information related to the use of their video system. From a literature review on video techniques, we found four studies providing details on the performance of their video systems: Pelletier et al. [26] for STAVIRO, Gladstone et al. [31], Santana-Garcon et al. [46] and Langlois et al. [47] for BRUVs. The size of the boat (small boat between 5 and 10 m), the number of persons required at sea (two persons minimum) and video analysis (one person assisted by experts as required) were common to all approaches. The number of stations sampled per day varied between studies (from 10 to 30 stations/day) depending on the number of systems used simultaneously, the duration of the footage and the distance between stations. Two to ten video systems were used per boat, with footage lasting from 9 min [26] to 180 min [46] and distance between stations varying from 200 to 500 m. The time required to analyze videos depends on the complexity of the habitat, as well as the diversity and abundance of fish. The analysis of RUV360 video was faster (49 min for a 25 min video on average, corresponding to 2 min per minute of video) than for STAVIRO video (43 min for a 9 min video on average, corresponding to 4 min 47 per minute of video) [26], because all fish present are visible within one frame, whereas six sectors of 60° are necessary for STAVIRO to get a 360° frame. The RUV360 takes longer to analyze than the BRUV (65 min for a 60 min video on average which correspond to 1 min per minute of video [31,46]) because fish are attracted to the camera with BRUV and are easier to identify, whereas greater zooming in is necessary with the RUV360 for species identification. The performance of the RUV360 was also linked to the nature of the videos analyzed, as soft bottom habitats are easier to analyze than complex habitats such as coral reefs.

### 4.2. Biodiversity Sampled

The RUV360 method was successful at sampling bare soft bottom habitats, seagrass beds, macroalgae meadows and mixed soft bottoms. The fish assemblages were significantly different according to the type of the soft bottom habitat. The differences were mainly driven by the presence of hard substrate, corroborating the observed relationship between the complexity of marine habitats and the composition of fish assemblages [2]. Structurally more diverse habitats are known to sustain fish communities which are more diverse and functionally complex in comparison with habitats with monotonous bare substrates [48]. The fish assemblages were first discriminated in the mixed soft bottoms (88% of correct classification), followed by vegetated or bare soft bottom habitats (59% of correct classification for each habitat). There were no clear boundaries between the vegetated soft bottom and the bare soft bottom assemblages, which form a continuum along a plant density gradient. Within the studied area, marine plants were common (only 3% of the stations had less than 10% plant cover). Therefore, even if bare soft bottom habitats were mainly composed of bare bottom, they also included marine plants to a lesser extent (<50%). The presence of marine plants on these habitats, and their associated species, can explain the difficulty of better discriminating fish assemblages between the vegetated and bare soft bottoms habitats. It appears that fish communities change along a gradient of marine plant abundance.

We recorded 10,007 fish belonging to 172 species (98 genera and 37 families), including 45 species (3365 individuals) targeted by fishing in New Caledonia and many emblematic species such as rays, sharks, turtles and dolphins; 104 sea snakes were also observed in the study area. For video analysis, several species were aggregated into groups, because they are similar in appearance and difficult to distinguish from each other. Grouping species which share specific traits in relation to their habitat, biology, behavior and ecology is common for studies using video techniques [31,46,49,50]. Another group of species seen in videos during this study could not be identified (26%), as they were too small or at the limit of the detectability (too far or too high in the water column from the camera). The observation of cryptic fish such as gobies (Gobiidae) and blennies (Blenniidae) is challenging using video, as they are too small and were often too far from the camera to be identified [21]. These two families represent a large number of species throughout New Caledonia (255 species on reefs and soft bottoms; [51] a number of the unidentified individuals in this study belonged to these two families. The difficulty of undertaking a census of cryptic fish species is not only related to the video analysis technique applied; it has also been reported in other, nonextractive sampling methods such as underwater visual censuses (e.g., [52]). Our results are consistent with previous knowledge of the biodiversity of lagoon soft bottoms in New Caledonia. Invertivores species dominate the assemblages, ahead of herbivores, piscivores and plankton feeders [1,11]. We observed 156 species out of the 542 species (28%) recorded on soft bottoms in New Caledonia using trawls or underwater visual census techniques (MK, pers. comm) [1,11,53]. The videos captured 16 additional species: 10 were hard bottom species observed on mixed soft bottoms, two were ubiquitous species, two were sharks and two were rays. The differences between videos and these other techniques are linked to the study area (location and size) and the techniques themselves. Bottom trawls census fewer hard bottom species because mixed soft bottoms cannot be trawled when the seafloor becomes too irregular, and most large species will avoid the trawl [53]. Video techniques are not adapted to census cryptic species [31,46,49,50]. Pelagic species are more frequently censused using UVCs than video techniques, and these species are seldom targeted by bottom trawls [54].

Consumer grade, spherical camera systems are significantly less expensive than high cost underwater cameras. However, the resolution may be sacrificed for the large field of view. Consequently, the range to which fish are identifiable will likely be reduced compared with high cost standard cameras, and this effect could be species-specific. Previous field tests using underwater benchmarks for distances indicated that we can identify fish at a typical maximum range of 8 m from the camera [47]. As species size also has an impact on detectability, we propose a list of species identifiable on the habitat sampled and have grouped similar species together.

### 4.3. Optimization of the Sampling Design Using RUV360

In order to optimize the sampling design (recording time per station and number of stations) using the RUV360, we had to collect representative, stable and reproducible data on soft bottom fish communities. During our study, 99% of the theoretical total species richness according to footage duration (="theoretical SR-time") was censused by the RUV360 in the area, using footage of 25 min. This demonstrates that it is not necessary to extend the duration of the footage, as 95% of the theoretical SR-time was observed within 14 min. The duration of footage varies greatly between studies, depending on the video technique used and the purpose of the study (from 8 min to several days) [21,33]. None of the studies referenced here specified the proportion of theoretical SR recorded, according to the duration of the footage taken. Therefore, subsequent results are strongly linked to the length of the selected footage. For example, according to the review on BRUVs by Whitmarsh et al. [33], 32% of BRUV studies used 60 min, 25% used 30 min and 17% used soak times greater than 90 min.

The RUV360 also recorded 88% of the theoretical species richness in the study area according to the station sampled, using nine stations/km$^2$. Very few studies using video

investigated the optimal number of stations required to obtain stable observations of biodiversity, and none of them reported this number in relation to the surface of the area studied. To the best of our knowledge, no experiments have investigated the impact of replicate spacing on observed assemblages ([33] for BRUV). For example, Santana-Garcon et al. [46] gave an optimal sample size of at least eight replicates per treatment in sampling a pelagic fish assemblage with a BRUV technique, while Gladstone et al. [31] concluded that for BRUV, there is no optimal value related to sampling precision, with values needing to be set by researchers according to the specific objective.

When designing a sampling strategy for soft bottom fish communities using the RUV360, it is possible to adapt footage duration and sampling effort. Therefore, it is possible to favor a strategy of either "short videos on many stations", or "long videos on a limited number of stations". Based on the data obtained in this study, we propose two reference plots to help in this process (Figure 7 and Supplementary Materials Figure S2 for reference tables). The choice will be a compromise between achieving acceptable precision, the variables and/or species of interest, and the need to manage costs [31,55].

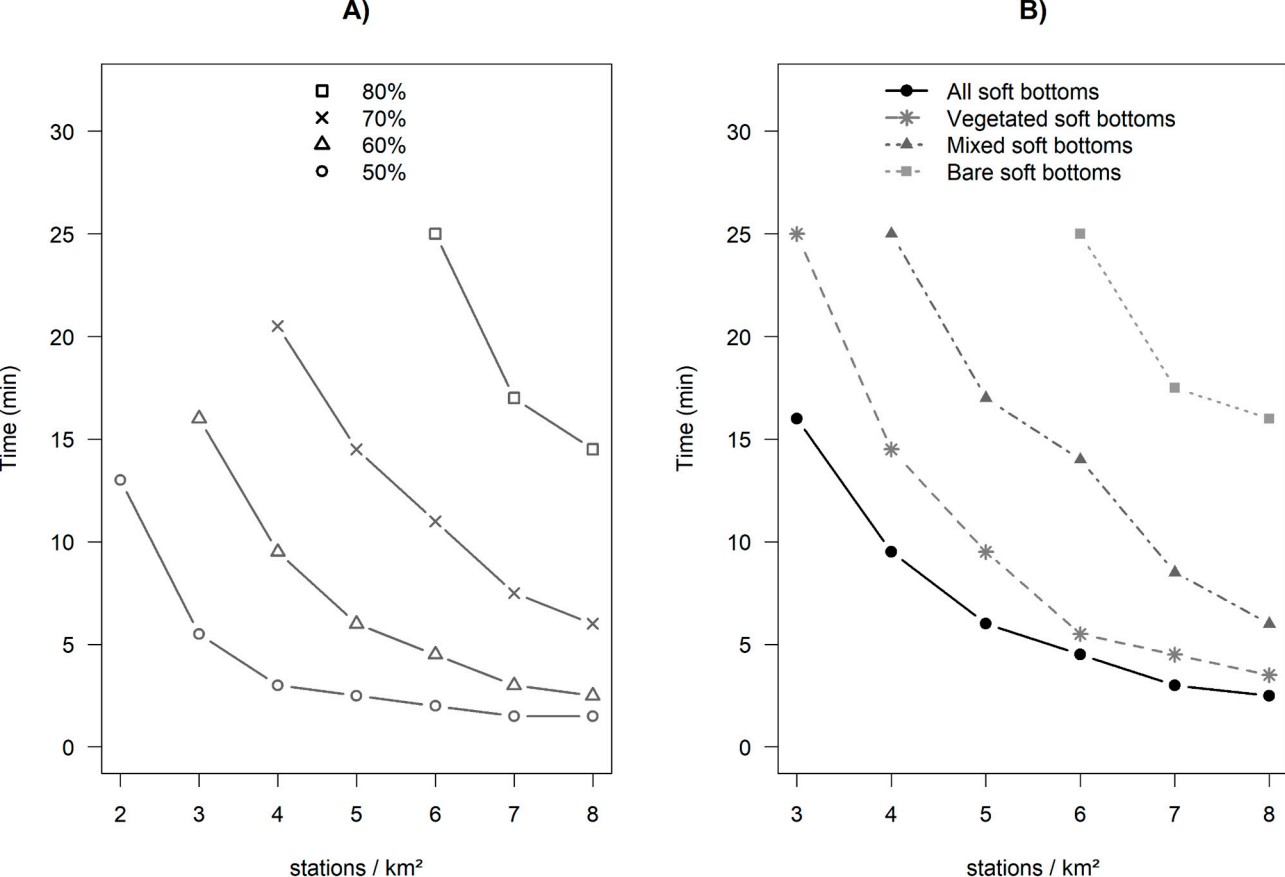

**Figure 7.** (**A**) Proportion of the theoretical SR-station depending on the duration and the number of video recorded within the studied area. (**B**) 60% of the theoretical SR-station per habitat depending on the duration and the number of video sampled.

## 5. Conclusions

The results of this study support the proposed sampling protocol to monitor fish communities in perireefal habitats during the daytime. To date, most attention in the scientific literature has focused on reefs, mangroves and seagrass habitats within the coral reef seascape. The sampling protocol described here offers the opportunity to obtain data on perireefal habitats that are comparable in space and time (specific richness, abundance) using a consumer grade 360° video camera. The results are consistent with the known characteristics of the lagoon soft bottoms fish assemblages, and the impacts of irregular

seafloors, current and visibility were limited. We provide reference plots to estimate the proportion of the theoretical total species richness sampled, according to the number of stations or the duration of the footage are provided. Further development should include the refinement of the method to collect body-size data from stereo video or other means. Body-size and length data are valuable for a range of ecological studies, from those focused on the impact of fishing to those on ontogenetic shifts of fish assemblages.

**Supplementary Materials:** The following are available online at https://www.mdpi.com/article/10.3390/fishes6040050/s1, Table S1: List of families and species of fish sampled in the study area. freq: frequency of occurrence; H1: primary habitat; H2: secondary habitat; S: soft bottom; H: hard-bottom; S/H: soft bottom and/or hard-bottom; C: Commercial fish: o (Wantiez pers. comm.). Figure S1: Mean cumulated number (±SE) of species and fish per station depending of the number of stations sampled. "MaxN": Abundance of fish; "SR": species richness. Figure S2. Proportion of the theoretical SR-station depending on the duration and the number of video recorded within the studied area for all soft bottoms and by habitats. See legend for colors.

**Author Contributions:** Conceptualization, D.M. and L.W.; data collection at sea, D.M. and M.O.; video analysis, D.M., M.O., S.I and L.W; data analysis: D.M., S.I. and L.W.; validation: D.M., M.O., S.I., M.K., L.W.; writing–original Draft Preparation, D.M., M.O., S.I., M.K., L.W.; Project Administration, D.M. and L.W.; Funding Acquisition, D.M. and L.W. All authors have read and agreed to the published version of the manuscript.

**Funding:** This research was funded by VISIOON, the Foundation of the University of New Caledonia, the University of New Caledonia, the Research Institute for Development (IRD), the Southern Province of New Caledonia and ADECAL-TECHNOPOLE.

**Institutional Review Board Statement:** Not applicable.

**Acknowledgments:** We are grateful to Jo Scutt Phillips from the Pacific Community (SPC) for helpful comments and English checking. We thank the Aquarium des lagons of Noumea for its logistical support during the development of the video system. We also sincerely thank the entire team of boat pilots from IRD Noumea, Miguel CLARQUE, Samuel TEREUA and Philippe NAUDIN for their professionalism, their competence and their assistance at sea allowing us to optimize the sampling and quality of the data collected.

**Conflicts of Interest:** The authors declare no conflict of interest.

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
