# Peer review of "Nondestructive Monitoring of Soft Bottom Fish and Habitats Using a Standardized, Remote and Unbaited 360° Video Sampling Method"

_fishes, doi:10.3390/fishes6040050_

Round 1

Reviewer 1 Report

General comments:

Very interesting paper with a good description of a potentially useful method.

Generally a very well written paper and the data analysis looks correctly carried out.

I have two major comments, that I hope the authors will introduce some discussion on.

Line. 343 “True representation” - that authors state that they used an unbaited method to estimate the “True” assemblage composition. The authors cannot state this as the very process of stoping a vessel on a point, dropping an unbaited camera to the bottom of the sea and then coming back to retrieve it again would produce many stimuli that many fish may respond to at various levels.

I suggest instead that the authors meant “Less biased representation..” and I suggest they use this or some similar language.

Body-size and length data. There are many published studies published now demonstrating the utility of body-size and length data for ecological studies of fish assemblages, from studies of the impacts of fishing to studies of ontogenetic shift. Many of the species found in this study are subject to both of the above processes. I recommend that the authors should at least add to the discussion that a further development of this method should be the refinement of it to collect body-size data, either from stereo video or some other means. 

Detailed comments:

Discussion  - points on efficiency in the field. The proposed system looks very cost effective. If the authors want more up to date information on logistics and number of samples per unit time they will find additional information in on stereo-BRUVs in Langlois et al. 2020. “A Field and Video Annotation Guide for Baited Remote Underwater Stereo‐video Surveys of Demersal Fish Assemblages.” Methods in Ecology and Evolution / British Ecological Society 11 (11): 1401–9.

Reviewer 2 Report

Major comments

The manuscript was overall very well conceived and written, and provides very useful data for optical survey design in soft-bottom nearshore habitat (maybe even beyond just coral reef lagoons).  The analysis was competent and fairly complete, addressing major issues that need to be addressed when starting a new survey time series, such as effort required.  I would have liked to see more on the final uncertainty in abundance indices for a few major species, and potentially some discussion on whether some type of stratification scheme (e.g.  abundance index by habitat strata) could potentially reduce index uncertainty. 

Also, the implementation of an off-the-shelf consumer grade spherical camera system is great, as often the high cost of underwater camera systems are substantial barriers for fisheries research groups.  I did wonder how the resolution is affected with this type of system, that is, is there an proximate measure of the spatial or angular coverage per pixel as there might be with standard view cameras?  I would imagine one would have to sacrifice resolution for the large field of view, and that targets occurring near the “edges” of the field of view may be most affected.  This would then potentially mean that the range to which fish are identifiable would be reduced compared with a standard camera, and this effect could be species-specific.  Perhaps the authors could add some discussion on this.

Minor comments

Line 108: Maybe rephrase "spherical lens with a 360 degree horizontal and 235 vertical view, pointed directly upward (fig 2)” for clarity

Line 123: “allowing for a clear view of the seabed”

Line 144: “eye color”

Line 146: Not sure why this paragraph is italicized

Line 152: “video analysis”

Line 169:  was “highly abundant” meant here?

Line 181:  Citation for Michaelis-Menten model

 Line 185:  t is defined twice in this sentence

Line 237:  this should be defined, is it the maxN summed across all deployments?

Line 244: mention that this is for the full 25 minute deployments

Line 276: odd font choice for “leather jacket”

Line 296: Is this theoretical SR time-accumulation model the Michaelis-Menten?  If so, indicate it here.

Line 483: “that are comparable”

Reviewer 3 Report

Dear Authors,

I found your manuscript very interesting, well written and organized. It explores a delicate topic such as sampling in biologically complex areas, using an innovative and especially non-invasive and non-destructive technique, suitable for those biological monitoring where it is not necessary even a collection of sample for further study, or which might in any case be associated with a form of animal sampling, if necessary.

The experimental design is appropriate and the results obtained are well shown and discussed, put in relation to the literature of the field, already rather rich, but that needed this further contribution, which makes, in my opinion, the manuscript worth publishing in Fishes Journal.

Despite the value of the manuscript, I found some minor points to address before publication, that I briefly report as follow.

Abstract

Line 23: I suggest to the Authors to evaluate the use of more common terms as key or representative, in place of emblematic.

Keywords

It is always good to avoid the use of terms already reported in Title between the Keywords (in your case: sampling method, fish, habitat). This is a good practice to give more resonance to a manuscript during the web research phases by other researchers.

Introduction

Line 37-43: this period is strictly related to study area, for this reason is more suitable for the related 2.1 section. I suggest to the Authors to replace this period with a more general one about this kind of environments.

Line 50: please better support this sentence with some related reference such as:

D’Iglio, C., Albano, M., Famulari, S., Savoca, S., Panarello, G., Di Paola, D., Perdichizzi, A., Rinelli, P., Lanteri, G., Spanò, N., Capillo, G. Intra- and interspecific variability among congeneric Pagellus otoliths (2021) Scientific Reports, 11 (1). DOI: 10.1038/s41598-021-95814-w

Brandner, J., Pander, J., Mueller, M., Cerwenka, A.F. and Geist, J. (2013), Effects of sampling techniques on population assessment of invasive round goby Neogobius melanostomus. J Fish Biol, 82: 2063-2079. https://doi.org/10.1111/jfb.12137

Results

Line 193-199: this period is really interesting, because focus the attention of the readers on some limit of the proposed method of the study. I found this kind of information really honest, realistic and constructive. Probably this subject should also be better deepened and supported in discussion and conclusions sections, for a better completeness of information for the reader.

Discussion

Line 342-346: please try to rewrite this period in a more clear form, it is quite confused.

Conclusion

Please deepen your conclusions by emphasizing the positive results, but also the limits that you yourself have highlighted in the results section.

In addition, the method should be related to daytime sampling, as the whole manuscript obviously excludes the possibility of using the proposed method at night, which other sampling techniques allow. This feature of the proposed sampling method should be better highlighted along the entire manuscript.

Have a good job.

Best regards

The Reviewer
